# Seeing the Error in My “*Bayes*”: A Quantified Degree of Belief Change Correlates with Children’s Pupillary Surprise Responses Following Explicit Predictions

**DOI:** 10.3390/e25020211

**Published:** 2023-01-21

**Authors:** Joseph Colantonio, Igor Bascandziev, Maria Theobald, Garvin Brod, Elizabeth Bonawitz

**Affiliations:** 1Graduate School of Education, Harvard University, Cambridge, MA 02138, USA; 2Psychology Department, Rutgers University, Newark, NJ 07102, USA; 3DIPF|Leibniz Institute for Research and Information in Education, Rostocker Str. 6, 60323 Frankfurt am Main, Germany

**Keywords:** Bayesian inference, cognitive development, learning, prediction, pupil dilation, science learning, surprise

## Abstract

Bayesian models allow us to investigate children’s belief revision alongside physiological states, such as “surprise”. Recent work finds that pupil dilation (or the “pupillary surprise response”) following expectancy violations is predictive of belief revision. How can probabilistic models inform the interpretations of “surprise”? Shannon Information considers the likelihood of an observed event, given prior beliefs, and suggests stronger surprise occurs following unlikely events. In contrast, Kullback–Leibler divergence considers the dissimilarity between prior beliefs and updated beliefs following observations—with greater surprise indicating more change between belief states to accommodate information. To assess these accounts under different learning contexts, we use Bayesian models that compare these computational measures of “surprise” to contexts where children are asked to either predict or evaluate the same evidence during a water displacement task. We find correlations between the computed Kullback–Leibler divergence and the children’s pupillometric responses only when the children actively make predictions, and no correlation between Shannon Information and pupillometry. This suggests that when children attend to their beliefs and make predictions, pupillary responses may signal the degree of divergence between a child’s current beliefs and the updated, more accommodating beliefs.

## 1. Introduction

It is not surprising that understanding the process of belief revision is of great interest and has a rich history in many fields, including philosophy, psychology, education, and computer science (e.g., [1,2,3,4]). Psychological and philosophical work suggests that two interrelated components of human intelligence are the ability to deploy abstract, causal, and “intuitive theories” to support inference and the ability to revise these theories in light of evidence [3,5,6]. Contemporary approaches in the Cognitive Sciences align empirical work with computational implementations, typically finding that Bayesian models can provide a framework with which to understand human inference from and learning of causal beliefs [7,8,9,10,11]. These models provide an account of how learners can draw rich inferences relatively rapidly even when data are limited or ambiguous and have been extended to account for the ways in which learners form and revise more abstract intuitive theories as well [12,13,14,15,16,17,18]. However, until recently, less work has investigated epistemic emotions and physiological expressions as they relate to rational models of human learning, despite the well-established connection between these arousal states and learning [19,20]. In fact, Bayesian models provide a means to not only understand how humans draw rich inferences from limited data and revise intuitive theories, but also to compare human physiological responses to competing computational theories of surprise and learning.

A large body of literature highlights the importance of affective and physiological states for learning and cognition in general. Physiological states, such as pupil dilation, are often accompanied by phenomenological affective states, such as surprise [21,22]. This is why many researchers studying the effects of surprise on cognition rely on objective physiological measures—such as pupil dilation—as a proxy for surprise [23,24,25,26,27,28]. However, *how* these physiological states relate to learning via belief revision remains less well understood. This challenge of determining *what* factors are closely linked to concept learning and *how* they affect learning is critical to address, as understanding these specific factors themselves provides multiple positive outcomes for research. Thus, conducting research computationally may improve our understanding of belief revision while also improving our ability to design human-inspired learning agents.

In the current study, we look to extend Bayesian learning models for investigating the potential relationships between the physio-emotional experience of surprise (as indexed by pupil dilation) and learning. Specifically, we contrast two predictive models related to learning: “Shannon Surprise” and “Kullback-Leibler divergence” belief updating. By building specific predictive models and relating them to children’s physiological responses (via pupil dilation), we can better understand the mechanisms that underlie learning in different contexts. Specifically, we investigate correlations between these two models and children’s pupillary surprise response as they perform belief revision during a water displacement learning task under different conditions. In one condition, children are asked to predict outcomes prior to observing events (engaging their prior beliefs), and in another, children make post hoc evaluations of the same evidence. By evaluating these different types of models and their fit to physiological behavior in these two conditions, we can better understand how different contexts might engage the interplay between cognitive and physiological mechanisms that support learning.

In what follows, we discuss the measure of pupil dilation and what pupil dilation indicates. Next, we describe scenarios where pupil dilation may most likely be elicited and more strongly linked to belief revision, namely when making predictions. Then, we investigate two candidates for computationally estimating the pupillary surprise response based on empirical findings and their theoretical interpretations. First, Shannon Information is investigated as a data-driven surprise response, and second, Kullback–Leibler divergence is investigated as a belief-driven surprise response. Thus, we aim to examine the specific challenge of understanding *how* pupil dilation response as a cognitive-behavioral response relates to learning via belief revision in our tasks.

### The Pupil Dilation Response, Attention, and Learning

Pupil dilation holds a special status among multiple connected fields, such as psychology, cognitive science, neuroscience, biology, and computer science. This is because pupil dilation has for a long time been considered a reliable instrument for identifying the temporal dynamics of arousal [29,30,31,32]. More recently, pupil dilation has been considered a physiological response that represents an integrated readout of an attentional network containing multiple contributing factors [33,34]. Within this attentional network, recent work suggests that pupil dilation in this network may occur as a result of an interactive cascade among various components, including low-level (e.g., light and focal distance; [35,36]), intermediate-level (e.g., alerting and orienting; [37,38,39]), and high-level factors (e.g., physio-emotional responses, inference, and executive function; [25,33,40]). Overall, accounts of pupil dilation as an attentional indicator highlight that pupillometry can broadly be attributed to either directed attention or higher-level sensory operations for processing the content that the observer is currently perceiving.

However, it remains unclear whether these discussed attentional factors and their related processes are what pupil dilation is expressing specifically in relation to learning. If so, whether some, none, or all of these factors are being expressed in the same fashion or to the same degree during belief revision. That is, we know quite a bit about what might elicit pupil dilation during learning scenarios (e.g., violations of expectations; [25,26,28,41]), but less about what the processes coupled with pupil dilation actually are and the implications of said processes. Thus, we propose designing computational Bayesian models of learning that can potentially estimate the degree of surprise experienced by learners, relative to their pupil dilation measurements, during a learning task.

With these Bayesian models, we will contrast two broader accounts of “surprise” that may help to clarify the relevance of this physiological marker of belief revision. The first candidate, originating from research on information theory (i.e., Shannon Information; [42]), posits that surprise (and thus, pupil dilation) correlates with objective expectations of the data and how informative it is given the data’s likelihood. The second candidate highlights divergence and dissimilarity (i.e., Kullback–Leibler divergence; [43]) between what is originally believed by a learner and what revised beliefs the learner expects to better accommodate the incoming data; it quantifies the degree of belief change needed to correctly represent the actual outcome of a given event by transforming the prior belief into an appropriate posterior belief.

In fact, recent work has looked into disentangling the pupillary surprise response as separable, distinct processes that can be represented computationally by Shannon Information and Kullback–Leibler divergence. One study by O’Reilly and colleagues [44] performed a combined brain imaging and pupillometry study where participants completed a saccadic eye movement response task. Here, the participants needed to use their prior knowledge about a spatial distribution to locate a target (a colored dot) before returning to a fixation cross. The findings showed that there were separate, specific neural signals associated with pupil dilation acting as temporal indicators of surprise (within the posterior parietal cortex) and belief revision (within the anterior cingulate cortex). Specifically, less likely events were considered more surprising via Shannon Information and elicited pupil dilation. Meanwhile, the authors found that the Kullback–Leibler divergence was related to when pupil diameters decreased on those trials when belief updating might be occurring. This work provides an important demonstration of the dissociable roles of Shannon Information and Kullback–Leibler divergence in computationally capturing surprise and belief updating, respectively, using a Bayesian framework.

Similarly, Kayhan et al. [45] investigated pupillary surprise and learning among 18-month-old infants and 24-month-old toddlers. Here, young children completed a statistical learning task that measured their pupil dilation as they viewed movies where an agent sampled five colored balls from a transparent bin containing multiple balls of two colors. These bins depicted the distribution of the ball colors inside of it (e.g., a majority of yellow balls (80%) and a minority of green balls (20%)). Critically, the 24-month-olds’ (but not the 18-month-olds’) pupillary responses followed a pattern similar to the prediction error of a causal Bayesian model, calculated as the Kullback–Leibler divergence between prior and updated probability distributions.

Thus, inspired by these exciting results, we designed a study that let us explore the further nuances of how different contexts (asking children to predict vs. post hoc evaluate outcomes) might engage the cognitive mechanisms associated with these two different accounts of surprise. This provides a means to explore the relationship between behavioral results that find differences in learning via different interactions with the physiological response and the potential cognitive mechanisms (surprise vs. belief updating) that might underlie them.

In what follows, we first describe these two potential mechanisms of pupil dilation and highlight key theoretical differences between their interpretations and implementations. Then, we describe specific contexts where these proposed mechanisms of pupil dilation may be most prevalent, via model-based prediction, as highlighted by a significant amount of recent empirical research. Next, we provide a brief description of the probabilistic Bayesian model used and what metrics we have investigated using it. Finally, we compare the two estimates of surprise—Shannon Information and Kullback–Leibler divergence—based on their correlations with children’s pupil dilation during a water displacement learning task.

## 2. Competing Accounts of Surprise: Shannon and Kullback–Leibler

### 2.1. Estimating Pupil Dilation as Data Driven via Shannon Information

Shannon Information is a well-known metric in information theory and describes how informative an outcome is [42,46,47,48,49]. It is largely found in machine learning literature to describe computational “surprise”—quantifying how meaningful incoming data are relative to a specific target despite other unwanted, noisy interferences. When interpreted with respect to learning (via Bayesian inference), Shannon Information can be used to describe the “unexpectedness” of incoming data, given the prior beliefs of the learner. Computationally, Shannon Information can be calculated as the negative log likelihood of some data’s probability, *p*(*d*), given some beliefs over models of the world (*H*), where Shannon Information Surprise (Equation (1)) is calculated as follows:(1)Shannon Information=−log(p(d)).

Shannon Information for some incoming data given an inferred model is typically quantified as a “signal” of information. Information theory captures this intuition as simply the negative log probability of the data. Note that this is computationally the same as marginalizing out hypotheses by considering the probability of the data given each hypothesis in space *H*, weighed by the prior probability of each hypothesis, *h*. One might interpret Shannon surprise psychologically as a violation of expectation, which depends on comparing the observation to a prior prediction of outcome likelihoods, given the weighted set of prior beliefs.

If Shannon Information correlates more strongly with children’s pupillometry compared to its competitor, the Kullback–Leibler divergence, then we posit that perhaps the pupillary surprise response may be more “objective” or “external-focused”, acting as a reaction to acknowledge the unexpectedness of an event that has occurred and draw attention to it. Specifically, “surprise as information” would represent an attentional mechanism homed in on the incoming data—emerging as a sign to alert the learner and re-orient (or heighten) their attention; this is a process of an “intermediate-level” of complexity among cognitive responses (according to a recent review of pupillometry research [34]). Thus, finding that Shannon Information best fits pupil responses may indicate a response akin to prediction error, which is typically associated with surprise during the violation of expectation events.

### 2.2. Estimating Pupil Dilation as Belief Driven via Kullback–Leibler Divergence

In contrast, other computational accounts describe pupil dilation and surprise in regard to how effectively the new data “transforms” a learner’s prior beliefs into their posterior beliefs [50,51]. Here, the summed Kullback–Leibler divergence is considered the second candidate for estimating surprise, measuring the summed dissimilarity or relative entropy between a learner’s distributions of prior and posterior beliefs, given the observation of some new data [43,52]. Computationally, the Kullback–Leibler divergence for models considering multiple, competing hypotheses is calculated (Equation (2)) as the weighted log-odds ratio between a posterior belief, *p*(*h*|*d*), and prior belief, *p*(*d*), which is summed across the hypotheses within the set of hypotheses considered (*hϵH*):(2)$$\;\;Kullback-Leibler\;Divergence=\overset{}{\underset{h\epsilon H}{\sum }}p\left(h|d\right)\log\left[\frac{p\left(h|d\right)}{p\left(h\right)}\right].$$

As mentioned, Kullback–Leibler divergence calculations describe not simply a distance between distributions, but also a measure of dissimilarity between them. Thus, when describing belief revision processes, Kullback–Leibler divergence can be considered as how much “work*”* is needed to affect an initial probability distribution (e.g., one’s prior beliefs) in a way that changes it into another (e.g., updated posterior beliefs). Here, if we find that Kullback–Leibler divergence relates to learning responses, then we believe that pupil dilation may be a more “subjective” physiological marker of learning that follows from the belief updating process. A symmetric (and finite value) form of Kullback–Leibler divergence (Jensen–Shannon) can also be used to similarly compute “distance”. In the computational analyses that follow, we apply the standard Kullback–Leibler divergence (in Equation (2)), but the results are not qualitatively different if the Jensen–Shannon divergence is used instead.

Central to our empirical question, this computational approach will allow us to contrast different models of “surprise” when learning. Specifically, Shannon Information quantifies the probability of the data accumulated by the learners trial by trial. Here, Shannon Information might be depicting pupil dilation as a temporal indicator of when children may be alerted to an unexpected, highly “informative” outcome that the child should orient themselves toward. Meanwhile, Kullback–Leibler divergence quantifies the dissimilarity between a child’s prior beliefs and what inferred models of the world would best explain the potential outcomes. This means that Kullback–Leibler divergence presents pupil dilation as a physiological signal of the amount of effort needed to update one’s beliefs (given the learner’s current belief distribution and the to-be posterior belief distribution that best explains the new data).

### 2.3. Model-Based Learning through Prediction

Asking learners to generate predictions is a popular method for improving children’s learning. Studies investigating prediction generation (or “hypothesis generation”) in children tend to find that explicitly predicting an outcome before seeing it improves learning (e.g., of physics; [53,54,55]). The benefits of making predictions have been connected to successful activation of prior knowledge when learning new material, but less is known about the specific mechanisms by which predicting affects learning success, in particular when it comes to theory revision [56]. Understanding the cognitive processes that are engaged during prediction generation can help us understand how, why, and when these interventions are likely to be successful.

Experiments that investigate pupil dilation and learning when making predictions find that actively generating a prediction compared to making a post hoc evaluation increases the degree of pupil dilation, particularly when faced with events that are incorrectly predicted [25,57]. Furthermore, previous work has found a positive relationship between the degree of pupil dilation and successful belief revision [27,28,40]. An enhanced pupillary surprise response after a violation of expectations may be due to children activating some task-relevant prior knowledge when they generate a prediction (i.e., leveraging their prior beliefs). Furthermore, if the outcome following a prediction is different from what the learner expects, then conflict awareness may be heightened and increase the subjective value of the outcome’s informativeness, which facilitates belief revision.

We hypothesize that, with all other things being equal, making a prediction may give children an “edge” over their peers and promote their learning by engaging the cognitive mechanisms associated with surprise. Assessing this prediction depends on two measures. First, it requires building models for individual learners that computationally predict when surprise is highest, given the learners’ beliefs and the observed evidence. Relating these model predictions to physiological markers, such pupillometry, helps us understand the computational and potentially mechanistic basis for pupil-marked surprise in learning. It also allows us to contrast competing computational markers of surprise under different learning contexts. Second, we can relate the degree to which individual children’s physiological states are correlated with these quantitative models and predict that children who have better “alignment” between the physiological and model-based surprise may also be more “optimal” learners, in the sense that their learning behavior is better matched to the idealized learning models. That is, if a heightened, “rational surprise” response leads to more efficient learning, then children who experience surprise when a rational model (e.g., a probabilistic Bayesian model) would expect them to may also be better simulated by said rational model as well. Thus, further investigation via computational models of these potential individual differences may reveal whether children whose pupillometric measures are better fitted to the model estimates are also more strongly represented by the simulated behaviors of an ideal learner, as depicted by an Ideal Bayesian learning model [11].

## 3. The Current Study

As described, the main hypothesis of this paper is that children who engage more with a learning task by making predictions will have stronger correlations between their pupil dilation measurements and the model estimates of the pupillary surprise response, compared to their peers who only make post hoc evaluations (specifically regarding the modeled data to be described below). However, two non-exclusive outcomes are considered here regarding which of the model estimates fit the children’s pupil responses. Recent interpretations of prediction suggest that actively making a prediction entails leveraging one’s prior beliefs and extrapolating potential outcomes, given those beliefs (e.g., [11,27,28,40]). Both the Shannon Information and Kullback–Leibler divergence accounts are consistent with this proposal because they both leverage prior beliefs when making predictions. However, they differ in the mechanism and the (potential) implications of leveraging those beliefs. If the pupil measurements for children making predictions are better matched by the Shannon Information metric, then this suggests that pupil dilation may indicate more robust engagement with the feedback they receive. In particular, good performance of the Shannon Information estimate may represent children’s heightened attention to evidence that violates their beliefs (e.g., [47,48,49]). Such a heightened response could support later learning by increasing arousal and, thus, improve the encoding of surprising data, but the Shannon response does not reflect the learning in the moment. However, if the Kullback–Leibler divergence performs better than Shannon Information, we would find support for physiological responses capturing belief updating in the moment, suggesting that children may be performing an effortful computation that captures the degree of belief change. Critically, assessing the performance of these candidate metrics of quantifiable pupillary surprise—both in general and in competition with one another—helps us better understand the role of surprise during belief revision. Does surprise simply serve to guide attention to relevant outcomes? On the contrary, does it aid learners by highlighting their beliefs and informing their integration of new information?

We modeled the data from an experiment that investigated elementary school (six- to nine-year-old) children’s theories of water displacement for the current model (experimental procedure, data, and empirical results are those found in [40]. The children’s causal beliefs of water displacement were chosen as children frequently have the misconception that water displacement depends on the weight of an object or a combination of weight and size rather than on its size only (e.g., [58]), thus providing an appropriate domain for the investigation of variability across individual children’s beliefs, as well as their impact on children’s subsequent learning. Furthermore, previous work has modeled these experimental data for an investigation of children’s learning during a belief revision task [11] and found very strong fits between the “optimal” Bayesian learning and the children’s performance on the task.

The to-be-modeled experiment’s design in [40] entailed a Pretest phase, a Learning phase, and a Posttest phase. A total of 94 six- to nine-year-old children (M_Age_ = 8.00 years, SD_Age_ = 0.96; 46% female) participated in the experiment and were randomly assigned to one of two experimental conditions—a Prediction or a Postdiction Condition. Before beginning the Pretest phase, all children first viewed a familiarization clip of an experimenter demonstrating how water got displaced by pressing a sphere underwater, where, importantly, the experimenter stressed that the spheres throughout the experiment were assumed to be held underwater by a rigid pole. This familiarization was performed to avoid the chance that the children would evaluate buoyancy instead of water displacement.

On each trial, the children (regardless of assigned condition) were presented with two spheres of varied features (e.g., in size, material, and/or weight) that were side by side (see Figure 1 for a trial example). Then, the children stated which sphere they thought would displace the most water (between two identical containers). These judgments were assessed using a 5-point scale (e.g., (1 = *certainly the left sphere*, 2 = *maybe the left sphere*, 3 = *equal amounts of water for both*, 4 = *maybe the right sphere*, 5 = *certainly the right sphere*). During the Pretest and Posttest phases, the children did not see the outcomes of the trials to allow for a clean initial assessment of beliefs (prior to learning) and final learning. The children were only provided feedback during the Learning Phase of the experiment according to condition. The children in the Prediction condition were asked to provide a response *prior to seeing the outcome*; responses were given values from 1 to 5 and the children stated their expectation (and confidence) about which sphere displaced more water. In contrast, the children in the Postdiction condition first saw the results of the presented trial, and then they were asked to state what their expectations had been (*prior to the evidence*). The measures in this study and others reveal that children are honest about their responses in these postdiction conditions.

The children’s pupil dilation measurements were taken during the Learning phase for both conditions. Pupillometry was collected using an eye-tracking camera (Eye-Link 1000; SR Research, Osgoode, Ontario, Canada). A “Pupil Baseline Phase” was included 750 ms prior to the “Results Phase” (see Figure 1). This was done to allow for comparisons of children’s pupil size changes between the prediction and postdiction conditions. The duration (750 ms) was kept short to (a certain degree) prevent the children in the postdiction condition from generating a prediction as well (see [25,57]). Throughout the whole trial sequence, a fixation cross was presented to guide the children’s view to the center of the screen between the phases. Additional details regarding the lengths of each sub-phase of the Learning Phase trials can be found in Figure 1.

### 3.1. Bayesian Model of the Pupillary Surprise Response

The Ideal Bayesian learning model that we employed for our investigations builds on a recent investigation of individual differences in children’s belief revision (the Optimal Bayesian model described in [11]). Here, the Bayesian model constructs computational representations of the children’s beliefs based on their task responses. Doing so highlights the importance of individual differences in prior beliefs during learning, while further demonstrating the impact of multiple, competing beliefs that guide inferences, as the Bayesian model’s correlations to children’s behavioral responses are significantly stronger than competing frameworks for the entire subject pool (Bayesian Correlations > 0.8; Directional accuracy > 90%). Additionally, this model finds that the children in the experiment’s Prediction condition are better simulated by the model than the children in the Postdiction condition.

For the Ideal Bayesian learning model [11], a computation of the children’s prior beliefs is motivated by the findings from prior research that justify the characterization of three specific (“Size”, “Material”, and “Weight”) competing theories (and specifically basing children’s intuitions on just these three) when reasoning about water displacement, (e.g., [58]) with implications of how they may influence one another. Based on this theoretical foundation, three modeling stages are performed to mathematically represent the children’s belief states and the probability of an event occurring. The first modeling stage entails capturing representations of the children’s possible prior beliefs by evaluating the probability of the children holding each mental model, given their responses on the pretest. The second modeling stage captures the process by which these representations evolve throughout the learning phase of the experiment. The third modeling stage describes our method for computing the trial-by-trial Shannon Information (based on the probability of the outcomes on each trial, given each child’s individualized prior beliefs) and Kullback–Leibler divergence (based on the “dissimilarity” between a child’s prior beliefs and posterior beliefs for each trial). The details pertaining to Stages 1 and 2 can be found in Appendix A as well as [11]. Stage 3 is described below.

### 3.2. Estimating Children’s Trial-by-Trial Surprisal

We build upon the Bayesian model’s simulations for estimating the children’s pupil dilation measurements during the original experiment. Specifically, we look at the Bayesian trial-by-trial surprise predictions for individual children. The children’s estimated beliefs are based on their responses during the pretest and follow Bayesian posterior updating during the test trial observations (*p*(*h_t_*|*d_t_*) ∀ *h_t_ϵH_t_*). Surprisal (whether Shannon or Kullback–Leibler divergence) for each trial depends on an individual child’s expected belief state, given the evidence for that trial.

The children’s beliefs about how much water will be displaced by different objects have been identified in past literature (e.g., by Burbules and Linn, 1978 [58]), falling into relatively simple causal rules for predictions: a rule based on the size of the objects, one based on the material of the objects, one based on the mass of the objects (a mixture of size and material), and one reflecting random responding. Thus, in our model, the children’s beliefs are represented computationally as a distribution across these four possible beliefs (“Size” (*S*), “Material” (*M*), “Mass” (*W*), and finally “Random” (*R*)). Each child’s “model” (*p*(*H_t_*|*d_t_*); Equation (3)) of water displacement on a given trial (*t*) could be represented as the posterior probability over just four rules (S, M, W, and R):(3)$$p(H_{t}|d_{t})=[p(h_{st}=S|d_{t}),p(h_{mt}=M|d_{t}),p(h_{wt}=W|d_{t}),p(h_{rt}=R|d_{t})].$$

#### 3.2.1. Calculating Shannon Information

From Equation (1), we derive the model’s trial-by-trial *SI* surprise estimates in Equation (4). That is, on some trial (*t*), we determine the likelihood (*p*(*d_t_*|*H_t_*)) of that trial’s new data (*d_t_*) observed by the child, given their currently inferred model (*H_t_*):(4)$$SI=-log\left(p\left(d_{t}\right)\right).$$

Here, Equation (5) describes how our model calculates the probability of the data (*p*(*d_t_*)) on a given trial (*t*), as marginalizing over the four competing beliefs at time *t,* (*h_i =_ h_(w,m,w,r)_*)*,* which is the summation over the likelihood and prior for each model:(5)$$p\left(d_{t}\right)=\sum_{h_{,i\;}\epsilon \;H_{t}}^{}p(d_{t}|h_{t,i})p\left(h_{t,i}\right).$$

The likelihood of the evidence on trial *t*, *p*(*d_t_*|*h*) is computed for each hypothesis *h* (S, M, W, and R rules). The likelihood is then weighed by the strength of belief for each model *p*(*h*) under this summation. Thus, the evidence that is less likely under more strongly held beliefs will contribute more to surprise than when the evidence is unlikely under a weakly held belief (See Figure 2 for illustration).

#### 3.2.2. Calculating Kullback–Leibler Divergence

From Equation (2), we derive trial-by-trial Kullback–Leibler divergence as a surprise estimate in Equation (6). For some trial (*t*), we calculate the relative entropy for each considered belief (hypothesis *h_t,i_*) within the child’s currently held distribution of prior beliefs (*p*(*h_t,i_*|*d_t_*) ∀ *h_t,i_ϵH_t_*) with its respective posterior belief, *p*(*h_t + 1, i_*|*d_t + 1_*). Kullback–Leibler divergence (*KLD*) is taken as the sum of these relative entropies between the prior and the posterior beliefs, capturing the shift in distributions between time (*t*) and after observing the data at time (*t + 1*):(6)$$\;KLD(H_{t+1}||\;H_{t})=\overset{}{\underset{h_{t,i}\epsilon \;H_{t}}{\sum }}p(h_{t+1,\;i}|d_{t+1})\;log[\frac{p(h_{t+1,\;i}\;|d_{t+1})}{p(h_{t,\;i}\;|d_{t})}].$$

Here, on a trial (t), the data have not yet been observed and capture the distribution of the beliefs prior to observing the evidence, whereas trial t + 1 captures the posterior distribution. Kullback–Leibler divergence is simply capturing the relative change between the prior and the posterior beliefs, given some observations (See Figure 3 for illustration).

Both the Kullback–Leibler divergence and Shannon Entropy are calculated given a prior belief at time *t* for each child, for each trial. As noted in Section A.2, initial priors are computed independently for each child, given the responses the children provide in the Pretest phase. Because the test trials provide fixed evidence and the likelihood is weighed by this evidence, the Bayesian model has no free parameters. Thus, the Kullback–Leibler divergence and the Shannon Entropy that depend on these computations similarly have no free parameters. Assessing the performance of these candidate metrics of quantifiable pupillary surprise—both in general and in competition with one another—provides a means to explore the implications of different learning responses to the data at the individual level in a trial-by-trial manner. If the Shannon Information (SI) estimates better correlate with the children’s pupil dilation, then this may suggest that pupil dilation is an indicator of robust engagement with the incoming data, particularly when it is of low likelihood and is highly “informative”. If the Kullback–Leibler divergence (KLD) correlates more strongly with pupil dilation, then this may suggest that pupil dilation is an indicator of belief updating “in-the-moment”. Assessing these correlations under different contexts (prediction vs. postdiction) allows for an exploration of potentially different mechanisms engaged by different types of learning interventions.

## 4. Results

### 4.1. Assessing Fit of Model Estimates

The analyses performed for assessing each of the surprise estimates, Shannon Information and Kullback–Leibler divergence, use direct correlations between the model predictions of and the children’s pupil dilation responses recorded during the experiment. Bonferroni correction is performed where needed for conservative analyses and interpretation, with correlation *p*-values tested against a Bonferroni-corrected alpha (Condition [Prediction, Postdiction] × Estimate [*SI, KLD*], *α* = 0.05/4 = 0.0125. All correlations discussed in the Results section are additionally compiled in Table A1 for ease of comparison.

#### 4.1.1. Condition: Combined Analyses

When looking at the full dataset (2890 trials across 94 children), we found no significant correlation between either the Shannon Information (*r*(2889) = 0.01, *p* = 0.49) or the Kullback–Leibler divergence model (*r*(2889) = 0.02, *p* = 0.12) and the children’s pupillometric measurements. As noted, our primary question involves assessing the models while accounting for two different response modalities (prediction and postdiction) to assess the potential differences between these interventions.

#### 4.1.2. Condition: Separate Analyses

We first explored the differences in the children’s pupillometric response as related to Shannon Information. The Shannon Information estimate did not correlate with the pupillary response for either the Prediction (*r*(1437) = 0.03, *p* = 0.20) or Postdiction (*r*(1461) = 0.01, *p* = 0.62) condition. In contrast, exploring the differences in the children’s pupillometric response as related to the Kullback–Leibler divergence did reveal differences. The Kullback–Leibler divergence estimate was significantly correlated with the children’s pupillary response within the prediction condition (*r*(1437) = 0.07, *p* = 0.004 < *α*). There was no correlation between the pupillary response and Kullback–Leibler divergence for the children in the Postdiction condition (*r*(1461) = −0.003, *p* = 0.90). The difference between the strength of the Kullback–Leibler divergence and Shannon Information correlations within the Prediction condition was also significant, (*z* = 2.98, *p* = 0.0014); similar results were found for a bounded version of the Kullback–Leibler divergence measure, the Jensen–Shannon divergence [59,60]. These results can be found in Appendix B. The correlations between Kullback–Leibler divergence and pupillary response were also significantly different between the Prediction and Postdiction conditions (Fisher’s *r*-to-*z* transformation; *z* = 2.11, *p* = 0.0174).

#### 4.1.3. Exploratory Analysis with Data Subsets

Sources of noise, such as individual differences in prior beliefs, and an identified critical learning period (both highlighted in previous modeling work; [11]) might have affected the correlation between the model estimates and pupillary surprise. Therefore, we looked to control for two additional sources of noise in our data via follow-up analyses. First, not all of the children in the study were still “learners”, as a subset of the participants began the Learning Phase with the correct Size belief. Applying the same method as above, we looked at just the children who did not have beliefs based on the correct theory of water displacement at the beginning of the experiment (19 children had the correct theory already, leaving *n* = 75 of 94 children who began with an incorrect theory, approximately equally between conditions). Re-analyzing the data with this subset replicated the results above. There was no significant correlation between *Shannon Information* and the children’s pupillometry for this subset of “learners” (overall *r*(2259) = 0.02, *p* = 0.29; Prediction, *r*(1142) = 0.04, *p* = 0.11; Postdiction, *r*(1116) = 0.01, *p* = 0.54). Meanwhile, while the *Kullback–Leibler divergence* had no significant correlation with the entire “learner” subset (*r*(2259) = 0.036, *p* = 0.08), there were significantly stronger correlations between *Kullback–Leibler divergence* and the pupil dilation response for the learners within the Prediction condition (*r*(1142) = 0.08, *p* = 0.002 < *α*), compared to the Postdiction condition (*r*(1116) = −0.002, *p* = 0.93; comparing conditions: Fisher’s *r*-to-*z* transformation; *z* = 2.15, *p* = 0.0158). The *Kullback–Leibler divergence* did not have a significantly stronger correlation than *Shannon Information* for the “learners” in the Prediction condition (*z* = 0.99, *p* = 0.16) for this subset, as would be expected by the small sample size and the fact that the children with the correct theory would have predicted low surprise for the trials across the full study. None of these correlations were significant when looking at the subset of “already-knowers” (overall for *SI, r*(639) = −0.05, *p* = 0.20; for *KLD, r*(639) = −0.02, *p* = 0.58), and when looking between the Prediction (for *SI, r*(294) = −0.07, *p* = 0.19; for *KLD, r*(294) = 0.01, *p* = 0.79) and Postdiction conditions (for *SI, r*(344) = −0.03, *p* = 0.52; for *KLD, r*(344) < 0.01., *p* = 0.99).

Our second subset analysis explored only trials where “learning” was likely to take place. Previous modeling of the children’s learning over the course of the study revealed that most children converged onto the correct Size belief by trial 19 based on their choice behavior (where the 19th trial was the 75th percentile of when the children in the study seemed to have “learned” the Size belief according to the model; discussed in more detail in [11]). The sharp-then-plateaued learning rate was likely because the initial trials (*n* = 9) in the Learning Phase provided no differentiation between the competing belief models (Size, Material, and Mass). They were selected to be “congruent” with all theories and, thus, offered no “surprise” for any model or opportunity for learning. Following a handful of incongruent evidence (trials 10–19), the majority of children revised their beliefs and began responding consistently with the correct Size belief. This design (no conflicting evidence to support learning initially, nor learning after the correct beliefs are settled) might have artificially created “noise” in our pupillometric correlations. This is because variability in the responses on the pupillometric measures caused by other artifacts could temper with the correlations due to a relatively large number of trials where the *Shannon Information* and *Kullback–Leibler divergence* estimates were both very low. Thus, we also looked at “critical learning trials”—those that started with the first incongruent trial (trial 10, where data would be differentiated by the competing beliefs) and extended to trial 19 where almost all children (*n* = 74 of 94 children) had learned the correct belief (size dictates water displacement) as measured by Bayes Posterior Odds. For these “critical learning trials”, we again replicated the overall pattern of results. The *Shannon Information* did not correlate overall during these critical trials (*r*(858) = 0.04, *p* = 0.24), nor did it correlate within either condition (Prediction condition: *r*(431) = 0.05, *p* = 0.26; Postdiction condition: *r*(426) = 0.06, *p* = 0.18). Again, the *Kullback–Leibler divergence* did not correlate for all children across all of the “critical” trials, (*r*(858) = 0.04, *p* = 0.24). However, (replicating the other analyses) there was a significant correlation between the *Kullback–Leibler divergence* estimate and pupillary response within the Prediction condition (*r*(431) = 0.12, *p* = 0.013 < *α*), while no correlation was found in the Postdiction condition (*r*(426) = −0.003, *p* = 0.90). These correlations were significantly different between the Prediction and Postdiction conditions for *Kullback–Leibler divergence* (Fisher’s *r*-to-*z* transformation; *z* = 2.11, *p* = 0.0174). The difference between the *Kullback–Leibler divergence* and *Shannon Information* for the Prediction condition yielded a significant difference as well (*z* = 2.98, *p* = 0.0014) during these “critical” trials. This suggests that the pupillary surprise response reflects something like belief updating, but only in conditions when children are actively engaged in prediction (a point we will return to in the Discussion).

### 4.2. Modeling Individual Differences

We were also interested in relating pupillary response and modeled surprise to learning. Thus, we looked at how, at the individual level, the degree of fit between the physiological response and the model response related to the degree to which the children’s responses reflected Bayesian “optimal” learning. That is, we correlated two correlations. Specifically, for this investigation, we looked at the correlation between the children’s behavioral answers (1–5) and the Bayesian model predictions of those answers as one set of correlations, and the children’s pupillary response performance and our models of surprise as the second set of correlations. If the pupillary response related to learning, we might expect to see that those children whose pupillary responses were more aligned with the model predictions were also the same children who learned more “optimally”. Indeed, we found that the correlation of individual children’s pupil response to the Kullback–Leibler divergence correlated significantly with the correlation of those children’s answers and ideal Bayesian learning (*r*(88) = 0.27, *p* = 0.007). In contrast, the correlations based on the children’s pupil response and Shannon Information did not correlate to this learning measure (*r*(88) = 0.05, *p* = 0.58). The difference between the correlation coefficients was marginally significant (*z* = 1.49, *p* = 0.06).

### 4.3. Understanding Differences between Shannon Information and Kullback–Leibler Divergence

Overall, we found that Kullback–Leibler divergence best aligned with the children’s behaviors in the Prediction condition, and that Shannon Information measures did not align with the children’s responses in either condition. This result might seem surprising given that, overall, Shannon Information and Kullback–Leibler Divergence might be predicted to significantly overlap. Indeed, the exploratory analysis revealed a significant correlation between both measures within the entire dataset (*r*(2988) = 0.38, *p* < 10^−97^), and separately within the Prediction (*r*(1436) = 0.37, *p* < 10^−47^) and Postdiction (*r*(1460) = 0.38, *p* < 10^−51^) conditions. However, there exist instances where the computed Shannon Information and Kullback–Leibler Divergence make different predictions (e.g., one example shown by comparing the “Material-” and “Mass-Dominant” belief distributions in Figure 2 and 3). Despite being calculated for the same child on the same trial (and thus being based on the same data), Shannon Information might predict “higher” surprise than Kullback–Leibler Divergence, or vice-versa. The exploratory analysis of our data revealed the strongest divergence between the predictions of these two models for trial events that provided belief-disambiguating evidence to children who were still transitioning between models. Whether Shannon Information was relatively greater than Kullback–Leibler divergence, or vice versa depended on the shape of the children’s beliefs (because Shannon Information and Kullback–Leibler divergence operate over different ranges of possible values, the exploratory analysis reported here compares values that are normalized to a 0–1 range within each approach). Specifically, when a child had a very strong belief in the Mass model (nearly zero weight across the other models), and a mild conflict trial was presented, Shannon Information predicted higher surprise and Kullback–Leibler divergence predicted lower surprise due to the overwhelming pull of the prior beliefs, limiting a distributional shift. In contrast, when the children’s beliefs were slightly more evenly distributed (especially between Mass and Size models), Shannon Information predicted lower surprise and Kullback–Leibler divergence predicted higher surprise. The children’s pupillometric scores best aligned with the Kullback–Leibler divergence for these models and the evidence-contrasting trials in the Prediction Condition in particular, further supporting the idea that the children’s eye dilation was related to key moments of engaged learning triggered by generating predictions.

## 5. Discussion

This paper describes one of the first computational investigations of the links between children’s pupillary surprise response and their science concept learning, as related to the contextual effects of engaging in an explicit prediction or postdiction. We modeled data, including pupillometric responses, collected from elementary school children who provided predictions or postdictions in a water displacement learning task. By modeling individual children’s beliefs and learning over trials, we could capture two different forms of “surprise”: Shannon Information and Kullback–Leibler divergence. Overall, we find that the children’s pupillary surprise response is related to the Kullback–Leibler divergence, but only in cases where children have generated an explicit prediction prior to observing the potentially surprising events. Furthermore, given the details of how the Ideal Bayesian model performs, it should be clear that both the Shannon Information and the Kullback–Leibler Divergence rely on children’s prior beliefs (and subsequent posterior beliefs) for their computations. The Shannon Information is computed based on the likelihood of an event, given some set of prior beliefs at the moment of the observed evidence, while the Kullback–Leibler divergence is calculated as the dissimilarity between the children’s prior and posterior beliefs when facing the same event. This difference is important to consider, as we found that children whose pupillometric data are best estimated by the Kullback–Leibler divergence also tend to be the children whose behavioral response data (from an experiment on learning water displacement via belief revision) are best fitted to an ideal Bayesian learning model, but that this same correlation is not found for the children’s computed Shannon Information. This further supports the notions that pupil dilation may be linked to “higher-level” mechanisms being engaged during learning in our task, as compared to “intermediate-level” factors typically associated with Shannon Information. We return to this point further below.

Our findings fit well with the theory described at the intersection of cognitive, emotional, and physiological research (e.g., [33,34]), with particular links to recent work investigating the role of prediction in belief revision (e.g., [25,27,40,57,61,62]). Our findings also converge with other related research. Like Kayhan and colleagues [45], we found a relationship between pupil dilation and the Kullback–Leibler divergence. Both this previous work and the current investigation find that the calculated divergence may affect belief revision in regard to the amount of updating needed to adjust current beliefs. However, there are two key differences between our modeling work and that of Kayhan et al. [45] which are important to note. First, the current paper investigates children’s pupillary surprise under different contextual conditions. The current results find that the relationship between the modeled surprise (via Kullback–Leibler divergence) and the children’s pupillary surprise response may *only* occur when the children are actively making predictions—but not when they are passively observing and evaluating. This highlights that there are instances where pupillary surprise might be more likely to occur when making predictions, as proposed to the results of other recent empirical works (e.g., [28]). Second, in line with the original paper that we draw our model from [11], the current model accounts for individual differences among the children’s prior beliefs and the processes by which they update. In the study by Kayhan et al. [45], children’s behaviors are modeled to all follow the same inferred computational model. Understandably, we acknowledge the limitations of Kayhan et al.’s [45] investigation given the population being studied. Specifically, Kayhan and colleagues faced the challenge of investigating this domain in 18-month-old infants and 24-month-old toddlers. Thus, acquiring explicit measures to inform computational representations of prior beliefs might have been difficult or not plausible. In contrast, we formalize the prior beliefs that children may have at the individual level, as informed by their past behavior.

Like other work investigating surprise during learning, we found a relationship between the Kullback–Leibler divergence and the pupillary surprise response (e.g., [44,45]). However, unlike O’Reilly and colleagues [44], we did not find a relationship between likelihood-based Shannon Information and pupil dilation. One potential reason for this divergence is that there are differences in the degree of complexity of the learned “concept” of each study and in the number of hypotheses considered. Specifically, the previous work entailed a task that only required reasoning about one variable (the angle that the target appeared at on a screen; [44]); however the angle of the target might have taken many different values. In contrast, the currently modeled task might require reasoning about more complex, causal beliefs (e.g., whether an object’s size, material, or weight determines the amount of water displaced and how each of these features generates displacement; [40]), but only consider a few possible hypotheses. It is of course likely that the children were entertaining a more varied set of potential causal beliefs about displacement than the four considered here. The responses in the pretest aligned well across these four and previous work had focused on these, but we are open to there being a more complex space of beliefs in this domain as well. Indeed, as learners consider more complex interactions (e.g., buoyancy, water-permeable materials such as sponges), the space will balloon. Thus, one particular reason for the significant relationship in previous work between pupil dilation and Shannon Information (or likelihood-based prediction error), and the poorer fit with children’s pupillometry in the Prediction condition of the current dataset, may relate to either differences in the complexity of the concept being inferred or differences in the size of hypothesis space being considered.

A second difference between our results and the results of O’Reilly and colleagues’ [44] is that we found a positive correlation between pupil dilation and the Kullback–Leibler divergence during prediction, whereas a negative correlation was found in this past work. Our task differed in both the types of beliefs being considered and whether children were actively engaged in prediction. If beliefs are already engaged in this process (as they likely were for our participants following the explicit prediction prompt), then a relatively instantaneous pupillary growth response to the observed outcome is feasible. In our task, the number of options being considered and “simulated” by the children is bounded, with children only deciding among five options (really three directional outcomes). Additional analyses investigating a bounded divergence measure, the Jensen–Shannon divergence [59,60], were also performed and are described in Appendix B. Importantly, the Jensen–Shannon divergence performs almost identically to the Kullback–Leibler divergence in terms of its correlations with the children’s pupillometry. One possibility is that the positive dilation we observed in the prediction condition captured the amount of mental effort generated by explicitly considering the outcomes over more complex hypotheses. It has been suggested that when the necessary “work” appears unexpectedly “large”, more mental effort may be exerted to accommodate the new information (e.g., to reduce the “work”; [63,64]), and this is reflected by increases in children’s pupil dilation—similar to findings linking reduction of uncertainty to the presence of signals from neuromodulators (e.g., acetylcholine and norepinephrine; [65,66]). Of course, we do not have enough evidence to confirm that pupil dilation actually accompanies a more “effortful” mental process (e.g., like those found by [29,67]), only that the found correlations indicate a relationship between pupil dilation and the amount of “work” needed to update beliefs.

### 5.1. Understanding Potential Cognitive Mechanisms

Both the Shannon Information and Kullback–Leibler divergence accounts of pupillary surprise have support in the literature exploring cognitive mechanisms. Specifically, these proposed computational interpretations align with the attentional network described in previous work and are not necessarily exclusive. Shannon Information has been suggested to relate more to the “intermediate-level” factors, addressing what it is externally that a learner might be trying to process when pupil dilation occurs (e.g., [37,38,39]). Similarly, Kullback–Leibler divergence has been suggested to represent “higher-level” factors relating to internal processes and state-like fluctuations that the learner might be experiencing (e.g., [25,33,40]). Thus, support for either the Shannon Information or the Kullback–Leibler divergence accounts (or potentially both) in estimating children’s pupillometry would have fit with various findings and interpretations of pupil dilation as some form of attentional network activation (for a thorough review, see [34]).

If these accounts of Shannon capturing “intermediate-level” factors and Kullback–Leibler divergence capturing “higher-level” features are correct, our results provide support for “higher-level” factors being engaged in our task—at least when children are explicitly making predictions. Perhaps when making predictions, children are orienting their attention toward their beliefs. That is, pupil dilation in our task may be an indicator of children’s online assessment of their current models of the world and what the implications would be (how much effort is needed to change these models), given the potential outcomes of an upcoming event.

Why might Kullback–Leibler divergence capture greater attention or cognitive effort? As described earlier, Shannon Information quantifies a single signal of data informativeness against only the current hypothesis space [42,46,47,48,49]. In contrast, Kullback–Leibler requires a computation over two hypothesis spaces—the prior and the posterior beliefs. In this way, Kullback–Leibler divergence might reflect more effortful cognitive processes.

### 5.2. Limitations and Future Work

The implications of this work highlight key investigations that future work should pursue. Specifically, one such avenue entails empirically and computationally capturing a “construct” of surprise that accounts for its emotional, cognitive, and physiological components. Next, future work may also be interested in further refining our understanding of the “higher-level” processes that our results suggest as being associated with surprise—that is, the interactions between prediction, planning, and other executive functioning.

#### 5.2.1. The Noisiness of Pupillometric Measurements

We acknowledge the impact of noise within the original experiment’s pupillometric data, which could be due to many possible reasons. First, both the children and the model seemed to “quickly” learn the scientific concept (that size determines the amount of displaced water). Thus, opportunities for experiencing pupillary surprise might have been in short supply as the misconceptions of water displacement were not held onto for long. In response to this, we also analyzed subsets of the data to account for potential noise due to learning dynamics: whether children had already “known” the Size principle at Pretest, and the “critical” trials where learning would be most likely to happen. Doing so did lead to improvements in the fit between the Kullback–Leibler divergence when estimating surprise, but it did not affect the lack of fit with the Shannon Information. The second reason that noise might have been prevalent is that despite the best efforts to ensure careful task administration and data collection, there do exist drawbacks when collecting the pupillometric measures. For example, a careful preparation of the study’s location is needed. When collecting the modeled data ([40]), great efforts were made to prepare the study location at a local science museum. For example, the experimenters used a room with no windows, allowing only for artificial light to keep the light levels as consistent as possible, as low-level issues such as light levels and focal distance do affect fluctuations in pupil size [34,35,36]. This is important to acknowledge, as many interpretations of pupillometry entail an assessment of the average change in pupil size within a timeframe. Additionally, previous work investigating the influence of low-level factors, such as light levels, finds that pupil dilation can be oscillatory with respect to fluctuations in the luminance of objects and their environments [68]. This may lead to a pupillary surprise response with a short latency (relative to the measured timeframe), but particularly strong amplitude being washed out by the constriction of the pupil (whether by nervous system relaxation or slight light level variance) during the timeframe when the measures are averaged.

Finally, following the acknowledgment of the potential sources of noise, we also acknowledge the relative strength of the found correlations (e.g., in order of the Results section, the significant correlations showed Pearson’s correlation coefficients of *r* = 0.07, 0.08, 0.12). However, these correlations were found to be significant even when performing the analyses conservatively (via Bonferroni correction). To the best of our knowledge, this work seems to be the first to find significant correlations between pupillometry and a computational model estimate during science concept learning.

#### 5.2.2. Capturing Pupillary Surprise across Modalities

Notably, we found no correlations between either the Shannon Information or Kullback–Leibler divergence and the pupillometric measures of the children in the Postdiction condition. As described in the previous section, this might be partially due to noise leading to underpowered detection. However, it might also suggest that perhaps another mechanism (and thus another model surprise metric) needs to be considered and investigated in future work regarding when (or even, if) pupillary surprise occurs in different response modalities. The current work highlights that, when making predictions, pupil dilation may be indicating the performance of higher-level, learning-effort estimates. However, we did not find significant correlations between the pupillary response and model predictions in the postdiction condition despite the fact that, over the course of the experiment, these children also learned. Indeed, pupil dilation did occur at times during the original study for the children in the Postdiction condition—just not in a way that correlated with the models of surprise. Thus, future work should investigate whether other response modalities indicate that such processes are being performed when pupil dilation is elicited, with theory-based metrics for computationally estimating said pupillometry.

#### 5.2.3. Empirically Measuring Surprise

In contemporary work on surprise, the physiological measure of pupil dilation is commonly collected as a proxy or a marker that signals an individual’s experience of surprise (e.g., [22]). This tends to be proposed due to the occurrence of pupil dilation following a violation of one’s expectations—often inducing heightened attention, physiological arousal (e.g., the release of noradrenaline and norepinephrine), and increased activity in brain areas (e.g., within the brainstem) related to monitoring uncertainty [23,30,69]. However, as with most emotions, special care needs to be taken when discussing the measures and expressions of affective states. In particular, surprise has received considerable research attention since the mid-20th century, which still informs the theoretical concerns regarding what surprise actually is, and has connected the (less so recently) disparate fields that investigate surprise (see [19,20,70]). Importantly, these conceptualizations and implementations of surprise only relate to the physiological instances of surprise’s attentional capacities. Thus, future research that looks to finely define surprise not only in terms of its proposed physiological markers but also its subjective experiential phenomena could also collect self-reported measures of experienced surprise as an additional correlate to further substantiate claims surrounding the physiological measures of surprise.

#### 5.2.4. Investigating Modalities That Potentially Leverage Prior Beliefs

Future work may want to consider investigating science concept learning by revisiting interview methods of past studies to further understand children’s subjective prior beliefs and what processes children (propose that they) may have employed to revise them (e.g., as in earlier water displacement studies; [58]). In fact, recent work highlights that thought experiments—imagining outcomes of an event and revising assumptions—can be beneficial for learning in both adults [71] and young children (six-year-olds; [72]). Thus, future work may tackle the integration of key experimental design aspects from the currently modeled data (the role of prediction and pupillometry) and research on other learning-by-thinking methods, such as thought experiments. Doing so may help determine whether such planning is being implemented by children. However, such approaches should be conducted carefully and interpreted cautiously, as meta-cognitive awareness and performance of thought experiments may be difficult to investigate, and work that focuses explicitly on whether people (especially children) typically benefit from thought experiments (compared to original work relying on allusions to scientific revolutionaries such as those made by Galileo, Kepler, and Einstein) is relatively new to the field [73].

#### 5.2.5. Potential Roles of Executive Function

Recent advances in research on attention highlight that top-down regulation and executive control are vital for processing and awareness of relevant information in the environment (extensively reviewed in [33,34]). Specifically, executive function is important for the guidance of intermediate-level attentional processes (e.g., alerting and orienting) for sensory operations. Here, we propose that future work should perform further computational investigations centered on incorporating measures of executive function. Modeling the relationships among theory change, prediction, pupillary response, and executive function skills (such as inhibition and cognitive flexibility; [74,75]) may provide further insight into other relevant mechanisms that support science concept learning. Such modeling would highlight whether executive function affects model performance straightforwardly, where higher executive function measures might correlate with better model performance. Additionally, future work may entail the design of Bayesian models that account for various executive function skills. For example, would a model that has the ability to inhibit incorrect prior beliefs perform better? Moreover, would a model that flexibly switches focus toward updated, “more correct” theories be plausible and sufficiently capture children’s behavior?

## 6. Conclusions

Here, we have identified a candidate computational measure that may capture the pupillary surprise response in a quantifiable way when children are making predictions during science learning. Specifically, we found that when children make predictions, their pupil dilation in response to the observed outcomes may be a temporal indicator of the children leveraging their initial prior beliefs and extrapolating the implications of those outcomes, given said prior beliefs. The current work contributes to our knowledge of what pupil dilation may be an expression of during the learning process. Specifically, by identifying contexts where pupillometry can be estimated computationally via the Kullback–Leibler divergence, we have also identified candidate mechanisms and processes that children may be performing when pupil dilation is elicited. That is, since the Kullback–Leibler divergence typically describes dissimilarity, or the amount of “work” needed to transform one probability distribution into another, the current findings have highlighted that explicit prediction may elicit the pupil dilation response as a physiological marker of children’s belief revision—estimating how much “work” is needed to move from the prior to the posterior beliefs. This behavior was not found for children who were only post hoc evaluating, suggesting a privileged role for prediction in engaging learning-relevant physiological responses. This computational modeling investigation, alongside recent experiments centered on prediction, provides some initial insight into why engaging children to generate predictions may support learning more effectively than other interventions. Such a simple manipulation may differently engage affective states and impact children’s learning; that is perhaps most surprising of all.

## Figures and Tables

**Figure 1 entropy-25-00211-f001:**
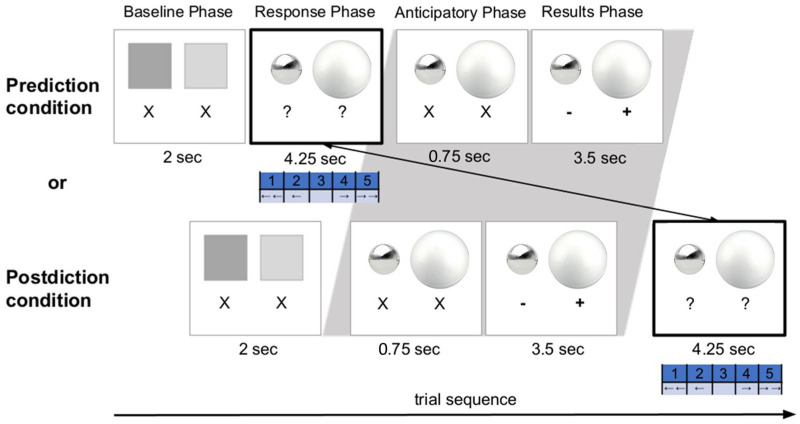
An overview of a trial’s procedure during the original experiment (Theobald and Brod, 2021). Children were randomly assigned to one of two experimental conditions (between-subjects): the Prediction or Postdiction condition, which differed in the timing of the children’s response (gray background) and the timing of related feedback. Specifically, the children in the Prediction condition first provided a response—stating their expectation about which sphere displaced more water—before the results of the event and related feedback were presented to them. In contrast, the children in the Postdiction condition first saw the results of the presented trial, and then they stated their expectations by providing an evaluation. Here, the correct response for the trial example is option “5—Right Wins”, noted by the “+” symbol under the Results Phase columns. However, this feedback is provided as evidence to support learning either following the children’s response (Prediction Condition) or preceding their response (Postdiction Condition). Children with the correct “Size” rule would have accurately selected “5” (or “4”) here and seen the confirming feedback. However, because, in this trial, the metal ball is made of a much heavier material than the styrofoam ball, despite its smaller size, children with the incorrect Material or Mass beliefs may incorrectly respond 1, 2, or 3 in their predictions or postdictions, and potentially be surprised by the evidence (that 5 “wins”).

**Figure 2 entropy-25-00211-f002:**
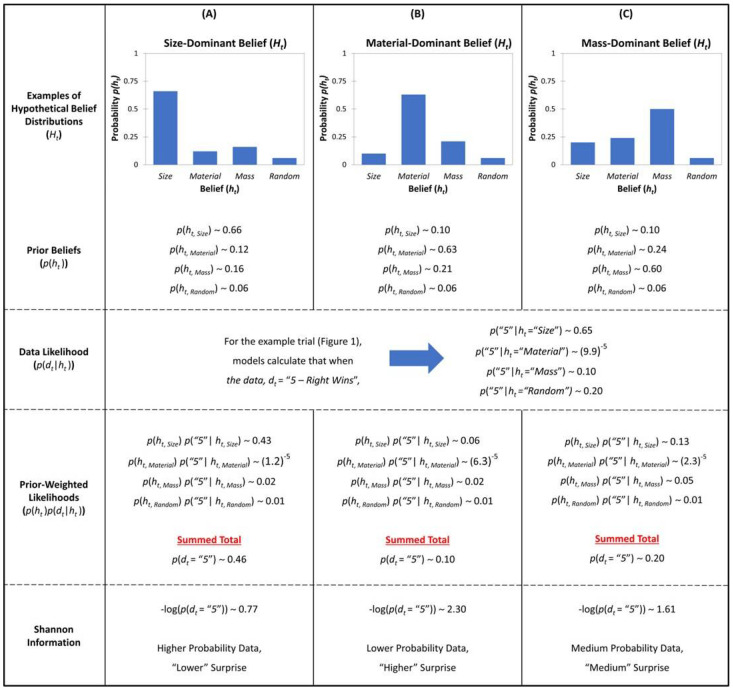
Example of the procedure for calculating Shannon Information, given the current model’s simulations, as formally described by Equation (4) and Equation (5). Columns (**A**–**C**) display three different examples of the typical kinds of profiles of children’s prior beliefs captured in graph and numeric form (Row 1). Given some incoming data (e.g., the example trial from Figure 1; a Small Metal ball vs. a Large Styrofoam ball), the likelihood of the observation (that event “5—Right Wins” occurs) is estimated for all four models (Row 2). Then, a posterior probability is calculated by weighing the individual child’s prior beliefs against the likelihood (Row 3). Shannon Information is calculated by summing over (marginalizing out *h*_t,i_) these posteriors and taking the negative log likelihood of the final summed total. Thus, there is an inferred negative relationship between data likelihood (*p*(*d_t_*) and model surprise according to the Shannon Information account (Row 4). That is, when the weighted likelihood of data is low, the model surprise is high; similarly, when the likelihood of data is high, the model surprise is low.

**Figure 3 entropy-25-00211-f003:**
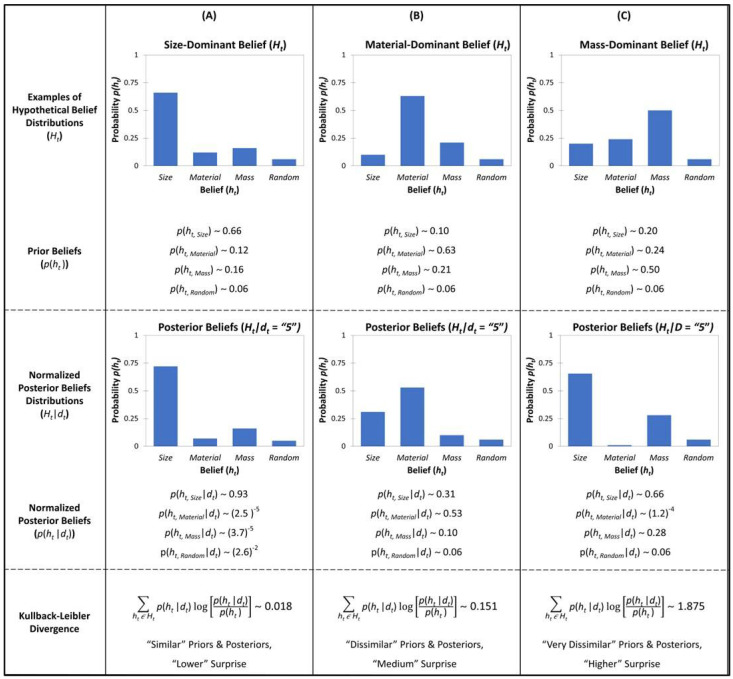
Example of the procedure for calculating Kullback–Leibler Divergence (KLD), given the current model’s simulations, as formally described by Equation (6). Columns (**A**–**C**) display three different examples of the kinds of typical profiles of children’s prior beliefs captured in graph and numeric form (Row 1). Given some incoming data (e.g., observing option “5” = right side wins for the example trial in Figure 1; a Small Metal ball vs. a Large Styrofoam ball) and the prior beliefs of the learner (Belief Distributions, *H_t_*), we consider the posterior belief distribution that best accommodates the observed data (e.g., *p*(*H_t_*|“5”)), which is, again, captured in graph and numeric form (Row 2). Then, the Kullback–Leibler divergence (Row 3) is calculated as the sum of relative entropies between the prior probability and the posterior probability between each of the individual competing beliefs (*h*_t,i_). Thus, there is an inferred positive relationship between the degree of dissimilarity between distributions (divergence between the prior beliefs and the posterior beliefs) and model surprise according to the Kullback–Leibler divergence account. That is, when the prior and the posterior beliefs are dissimilar, the model surprise is high; conversely, when the prior and the posterior beliefs are similar, the model surprise is low.

## Data Availability

The experimental data and code used in this study are available online at https://osf.io/wqnf2/?view_only=fba98d3304bc4a759e4346eaa7df3332 (accessed on 4 November 2022).

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
