# Peer review of "Seeing the Error in My “Bayes”: A Quantified Degree of Belief Change Correlates with Children’s Pupillary Surprise Responses Following Explicit Predictions"

_entropy, 2023, doi:10.3390/e25020211_

Round 1

Reviewer 1 Report

The paper investigates the link among children's pupillary surprise response and belief updating. Specifically, two mechanisms are discussed. Shannon information, as a measure of surprise given a subjects' idea of the world and it discrepancy and Kullback-Leibler divergence as the distance among the prior and posterior distribution encapsulating the prior expectation of a fact about life and the later change in that fact, given data/observations on the world.
First off, the paper is nicely written and it pursues an interesting question. I'm not familiar with the field of pupillary surprise responses but it seemed that the authors provided sufficient background for me to understand what the underlying ideas and rationale are. I did get lost here and there and I was missing information in some sections.

I found the section about the experimental setup slightly confusing (p. 6, lines 289 -- 306). I had to read it multiple times to understand that there were two groups, one of which was asked to provide a statement _before_ seeing the results and the other group was asked to provide the statement about their believes _after_ seeing the results of the experiment. Maybe pulling the sentence starting with "Children in the..." on line 300 up to the beginning of the paragraph would make things clearer.

There's many hypotheses in this paper:
On p6, line 247 the authors mention "...the second hypothesis..." I'm left wondering what the first hypothesis is?
Shortly after, a new hypothesis is introduced as "the broad hypothesis.." and just after that, two alternative "hypotheses". I don't think these last two hypotheses qualify as such, as they are not falsifiable: Depending on the outcome of the experiments either hypothesis 1 is selected or the other hypothesis 2 but there is no falsifiable argument in there.

The authors mention an "ideal Bayesian model" described in a currently unpublished manuscript. Please provide that model in the text.

The KL concept is a bit tricky, as it involves assigning prior probabilities to children's hypotheses at each trial p(h_ti). I don't understand how the prior belief was obtained? I would think that the prior influences the outcome of the analysis decisively, how robust are these findings to slight variations in those individual priors?

Is Figure 2 a representation of two prototypical children?

Maybe a short paragraph describing the sample would be helpful with total N, ages, sex distribution etc.

Lines 506--509: "we found that children whose pupillometry data was best estimated by the Kullback-Leibler divergence also tended to be the children whose behavioral response data was best fit by an ideal Bayesian learning model." I'm not sure what to make of this? It remains unclear what the "ideal Bayesian learning model" is but given that the KL incorporates the posterior and the prior it's probably not surprising that KL works well. This sounds circular.

While minor, I was wondering about the experimental item in Figure 1. While it is certainly correct that an object with a larger volume displaces more water than an object with smaller volume, it is easy to see why one would chose "left wins". Children will have made the experience that a styrofoam ball floats and hardly displaces water in that state while the metal ball will sink and displace its full volume. While this would not impact the "surprise" element it would not necessarily lead to better learning. I'm wondering if these items were chosen to be somewhat ambiguous on purpose or if this is unintentional?

Author Response

Please note that all Line and Page numbers refer to the formatted version of the manuscript. A version with changes tracked will also be uploaded with the revised version. Please see the attachment. 

Reviewer 2 Report

The authors present a secondary analysis of data from a previously published study. Here they correlate measures of pupil dilation with two Bayesian estimates of surprise, Shannon Information (SI) and Kullback-Leibler divergence (KLD). They show that KLD predicts a small but reliable proportion of the variance in pupil dilation, but only in conditions where participants make predictions about the correct answer during the learning phase (cf. postdiction judgments). The results are intriguing and should be of interest to those studying uncertainty in a range of learning contexts and/or pupillometry. I do have a few concerns which I hope the authors can satisfactorily address. Many are to do with lack of detail in the manuscript.

1)    The precise methods during the experiment are important for interpreting what these results might mean (especially since pupil dilation correlates with KLD in some conditions but not others). Even though this is based on an existing published study, I think the most relevant details need to be reiterated in the manuscript. It’s pretty vague at the moment. For instance, precisely when and for how long were the pupil dilation measurements taken? Perhaps I missed it but all I could find was this on lines 304-305: “Importantly, children’s pupil dilation measurements were collected as outcomes were presented during the Learning phase for both conditions.”

2)    Could the authors explain with a little more clarity where they got their p(dt|Ht) estimates? This is obviously very important for calculating the posteriors. For instance, refering to the examples in Figures 2&3, how do they arrive at p(“5”|ht = “Size”) ~ 0.65 (as opposed to any other relatively high probability) or that p(“5”|ht = “Material”) ~ (9.9)^-5 (as opposed to any other small probability). The description of how they estimated p(ht) from the pretest responses is also quite vague and could use some additional details.

3)    The examples worked through in Figures 2 and 3 are useful but they also highlight the fact that SI and KLD probably correlate fairly strongly (e.g. for this pair, a pair of hypothetical responders, A has lower surprise than B on both measures). Could the authors report what the relationship between SI and KLD is in the overall data set and in each of the constrained data sets?

4)    On the same note, could the authors also discuss the types of instances where SI and KLD make different predictions? It would be useful if the reader were given greater insight into why KLD provides a more informative metric than SI. It could be for the reasons the authors have stated but might also be related to the distribution of scores on the two measures. Or perhaps there are certain subsets of trials for which SI and KLD diverge and it is these trials where pupil dilation seems to track belief updating in particular?

5)    It is possible that the relationship information belief change and pupillary surprise varies in meaningful ways across individuals. If I understand it correctly the analysis presented here, which is not hierarchical, removes this level of information. Is there enough data available for each individual to get a sense of whether some individuals show stronger relationships than others?

Also, line 501: “provided predictions or predictions” – one should be postdictions?

Author Response

(The authors gave the same response as above.)

Round 2

Reviewer 2 Report

The authors have addressed my earlier concerns satisfactorily. I have a few very minor further comments below, but I don’t feel that I need to see the manuscript again.

New Footnote 5, page 14 — excellent that the authors ran exploratory analysis on normed data from the two indices, but what did this exploratory analysis indicate? Was there a similar pattern to the unnormed results? Or some divergence from the unnormed results (and if so, what)? Or are the main results reporting the results after norming? I think even one extra sentence of clarity in the footnote would help.

Appendix, line 935: “one based one the materials” — one should be on

Appendix, line 1001: “After generating the and posteriors of each trial” — word missing or extra word?

Author Response

Our responses to Reviewer 2's report is posted below, as well as in the attached pdf. Thank you for your time and consideration. 

___________________________

Reviewer: The authors have addressed my earlier concerns satisfactorily. I have a few very minor further comments below, but I don’t feel that I need to see the manuscript again.

Response: We are delighted to hear this and thank the reviewer for the thoughtful feedback which significantly improved the paper. In what follows, we will describe the changes made to the document to address the reviewer’s primary question and the listed minor proofing errors.

___________________________
Reviewer: New Footnote 5, page 14 — excellent that the authors ran exploratory analysis on normed data from the two indices, but what did this exploratory analysis indicate? Was there a similar pattern to the unnormed results? Or some divergence from the unnormed results (and if so, what)? Or are the main results reporting the results after norming? I think even one extra sentence of clarity in the footnote would help.

Response: Thank you for bringing this to our attention. We see that this footnote was ambiguous so we have reworded it for clarity. In short, there is no way to compare Shannon and KL Divergence scores directly because they have vastly different ranges. To provide a fair comparison, we rescaled (z-score) each measure so that the ranges were matched 0-1 prior to performing any analyses. The exploratory analysis reported here are based on these normed scores.

___________________________

Reviewer:

Appendix, line 935: “one based one the materials” — one should be on
Appendix, line 1001: “After generating the and posteriors of each trial” — word missing or extra word?"

Response: Thank you for your close reading of the revised manuscript. The mentioned typos have been corrected.
